# Insulin-like Growth Factor-1 and IGF Binding Proteins Predict All-Cause Mortality and Morbidity in Older Adults

**DOI:** 10.3390/cells9061368

**Published:** 2020-06-01

**Authors:** William B. Zhang, Sandra Aleksic, Tina Gao, Erica F. Weiss, Eleni Demetriou, Joe Verghese, Roee Holtzer, Nir Barzilai, Sofiya Milman

**Affiliations:** 1Department of Medicine, Division of Endocrinology, Institute for Aging Research, Albert Einstein College of Medicine, Bronx, NY 10461, USA; william.zhang@einsteinmed.org (W.B.Z.); saleksic@montefiore.org (S.A.); tina.gao@einsteinmed.org (T.G.); nir.barzilai@einsteinmed.org (N.B.); 2Department of Neurology, Albert Einstein College of Medicine, Bronx, NY 10461, USA; EWEISS@montefiore.org (E.F.W.); joe.verghese@einsteinmed.org (J.V.); roee.holtzer@yu.edu (R.H.); 3Ferkauf Graduate School of Psychology, Yeshiva University, New York, NY 10033, USA; eleni.demetriou@mail.yu.edu; 4Department of Medicine, Division of Geriatrics, Institute for Aging Research, Albert Einstein College of Medicine, Bronx, NY 10461, USA; 5Department of Genetics, Albert Einstein College of Medicine, Bronx, NY 10461, USA

**Keywords:** IGF-1, IGFBP-3, IGFBP-1, older adults, longevity, health-span, age-related disease, cognitive impairment, diabetes

## Abstract

While the growth hormone/insulin-like growth factor-1 (GH/IGF-1) pathway plays essential roles in growth and development, diminished signaling via this pathway in model organisms extends lifespan and health-span. In humans, circulating IGF-1 and IGF-binding proteins 3 and 1 (IGFBP-3 and 1), surrogate measures of GH/IGF-1 system activity, have not been consistently associated with morbidity and mortality. In a prospective cohort of independently-living older adults (*n* = 840, mean age 76.1 ± 6.8 years, 54.5% female, median follow-up 6.9 years), we evaluated the age- and sex-adjusted hazards for all-cause mortality and incident age-related diseases, including cardiovascular disease, diabetes, cancer, and multiple-domain cognitive impairment (MDCI), as predicted by baseline total serum IGF-1, IGF-1/IGFBP-3 molar ratio, IGFBP-3, and IGFBP-1 levels. All-cause mortality was positively associated with IGF-1/IGFBP-3 molar ratio (HR 1.28, 95% CI 1.05–1.57) and negatively with IGFBP-3 (HR 0.82, 95% CI 0.680–0.998). High serum IGF-1 predicted greater risk for MDCI (HR 1.56, 95% CI 1.08–2.26) and composite incident morbidity (HR 1.242, 95% CI 1.004–1.538), whereas high IGFBP-1 predicted lower risk for diabetes (HR 0.50, 95% CI 0.29–0.88). In conclusion, higher IGF-1 levels and bioavailability predicted mortality and morbidity risk, supporting the hypothesis that diminished GH/IGF-1 signaling may contribute to human longevity and health-span.

## 1. Introduction

The rise in age-related diseases and disability that accompany advanced age presents a burden to economies, health care systems, and individuals worldwide [1]. Evidence from model organisms demonstrates that aging is a biologically regulated process that can be modulated to extend lifespan and health-span [2]. The evolutionarily conserved growth hormone/insulin-like growth factor-1 (GH/IGF-1) pathway, which plays essential roles in growth, development, and metabolism, has been recognized as one key regulator of aging [3]. In invertebrates, attenuated signaling in pathways homologous to mammalian IGF-1 dramatically extends lifespan [4,5]. Similarly, mutant rodents with diminished GH and/or IGF-1 secretion or signaling exhibit lifespan extension of 25–60% [6,7,8,9,10]. They also display extended health-span, including delayed age-related impairments in cognition, musculoskeletal function, glucose homeostasis, immunosenescence and cancer [6,7,8,11,12,13,14]. On the other hand, overexpression of GH/IGF-1 in transgenic rodents accelerates age-related pathologies and dramatically reduces lifespan [15,16].

However, the role of the GH/IGF-1 system in human aging and longevity is still uncertain. In humans, profound reduction or enhancement in GH/IGF-1 signaling has consequences for survival and health that replicate some of those noted in experimental models. For instance, individuals with GH receptor deficiency are protected from lethal malignancies and type 2 diabetes, although their lifespan is not prolonged [17]. On the other hand, patients with acromegaly, characterized by hypersecretion of GH, have increased risk for premature cardiovascular disease, diabetes, malignancy, and mortality [18,19,20]. Nonetheless, epidemiologic studies investigating the relationship between circulating levels of IGF-1, which are used as a proxy for the activity of GH/IGF-1 axis in humans [21], and clinical outcomes have yielded inconsistent findings. While our group and others found inverse relationship between IGF-1 levels and survival [22,23], a number of studies reported positive [24,25] or null [26] associations. Furthermore, IGF-1 levels have been reported to have opposite effects on risk for different age-related diseases. For instance, lower levels of IGF-1 were associated with increased risk for cardiovascular disease [27,28], while high IGF-1 levels were related to increased risk for cancer [29]; although the findings were not consistent in all studies [30].

Several reasons may contribute to the inconsistent findings above. First, the activity of the GH/IGF-1 axis and levels of circulating IGF-1 are affected by acute [31] and chronic illness [32]; this introduces the possibility of “reverse causation” in some studies conducted in high-risk populations [25]. Second, numerous epidemiologic studies investigated associations between IGF-1 and morbidity and mortality in cohorts with wide age-ranges under the assumption that the effect of IGF-1 would be similar in younger and older adults [33,34]. Furthermore, total IGF-1 does not represent bioavailable IGF-1 [21]: Almost all circulating IGF-1 is bound to six IGF-binding proteins (IGFBP-1-6), leaving <1% of IGF-1 in a free form, bioavailable to bind to its receptors [35]. In addition to providing a long-lasting pool of circulating IGF-1, IGFBPs closely regulate biological functions of IGF-1 through controlled inhibition and promotion of IGF-1 interactions with its receptor [36]. Since measuring free IGF-1 remains challenging [37], the molar ratio of total IGF-1 to IGFBP-3, the most abundant IGFBP in circulation, is commonly used as a proxy for bioavailable IGF-1 [21]. Finally, other elements of the GH/IGF-1 system, including IGFBP-3 and IGFBP-1, have been implicated in human disease and survival, independent of IGF-1 [38,39]. Therefore, our aim was to prospectively investigate the associations between several components of the GH/IGF-1 pathway, including total IGF-1, IGF-1/IGFBP-3 molar ratio, IGFBP-3, and IGFBP-1, with mortality and incidence of major age-associated diseases in a cohort of independently-living older adults with majority in general good health at enrollment.

## 2. Materials and Methods

### 2.1. Human Cohort Data Acquisition

LonGenity is an ongoing longitudinal study initiated in 2008, that seeks to identify genotypes and phenotypes that protect from age-related diseases and promote exceptional longevity in humans [40]. The LonGenity cohort is composed of Ashkenazi Jewish older adults and about half of the cohort has a parental history of exceptional longevity, defined as having at least one parent survive to 95 years of age. Other inclusion criteria include baseline age ≥ 65 years or older and being free of significant cognitive impairment at baseline. Study participants are extensively characterized at annual visits, which include medical history and neurocognitive testing. Baseline IGF-1 and related protein measurements were available for 877 (54.5% female) study participants. Among this group, 37 individuals only completed the baseline visit and thus, were excluded from this analysis. Among the 840 participants included in this study, 20 did not have complete physical examinations as they either declined or were unable to do so due to mobility issues and were therefore missing body mass index (BMI) measurements. Fasting blood samples were also collected biannually at follow up visits. The LonGenity study was approved by the institutional review board (IRB) at the Albert Einstein College of Medicine. Informed consent was obtained from all study participants.

### 2.2. Biochemical Measurements

Biochemical measurements were performed as previously described [41]. Total IGF-1 levels were measured by liquid chromatography/mass spectrometry at Quest Diagnostics Nichols Institute laboratories (Quest, San Juan Capistrano, CA, USA) in serum collected at baseline and subsequently stored at −80 °C. For IGF-1, the limit of quantification (LOQ) was 15.6 ng/mL and the coefficient of variance (CV) was 3.3%, 3.1%, 2.8%, and 5% for the low (mean 57.2 ng/mL), medium (mean 248 ng/mL), high (mean 447.1 ng/mL) Bio-Rad quality controls and pooled human serum in-house control (mean 104 ng/mL), respectively. IGFBP-3 levels were measured at Quest with a chemiluminescent immuno-metric assay (Siemens Immulite 2000; Siemens Healthineers AG, Erlangen, Bavaria, Germany). IGFBP-1 levels were measured at Quest with a radioimmunoassay. For IGFBP-1, the LOQ was 5 ng/mL, and the CV was 9.3%, 10.1%, and 8.5% for the low (mean 19.2 ng/mL), medium (mean 53.5 ng/mL) and high (mean 111.3 ng/mL) controls, respectively. For IGFBP-3 the LOQ was 0.5 mg/L, and the CV was 5.1%, 6.1%, and 6.5% for the low (mean 0.90 mg/L), high (mean 3.56 mg/L), and in-house controls, respectively. The IGF-1/IGFBP-3 molar ratio was calculated by dividing measured serum total IGF-1 and IGFBP-3 levels by their molecular weights (7649 Daltons and 31,673 Daltons, respectively), and then calculating the ratio between the two quantities [42,43]. Insulin was measured by radioimmunoassay at the Albert Einstein College of Medicine Biomarker Analytic Research Core (BARC). Due to limitations in sample volume, IGFBP-3 was measured in 828 subjects, IGF-1 in 761, IGFBP-1 in 728 subjects, and insulin in 801 subjects.

### 2.3. Disease Definitions

Three of the age-associated morbidities in this study, cardiovascular disease, diabetes, and cancer, were selected because they represent major causes of morbidity and mortality in the aging population [44]. These morbidities were defined using a combination of self-reported questionnaire data, medical records, and laboratory results. Cardiovascular disease was defined as having a history of myocardial infarction, stroke, or cardiac procedure such as percutaneous coronary intervention or coronary artery bypass grafting surgery. Diabetes was defined as a self-reported history of diabetes, a fasting blood glucose of 126 mg/dL or greater, or a hemoglobin A1C level of 6.5% or greater. Cancer was defined as a self-reported history of any malignancy, excluding non-melanoma skin cancers.

We also investigated multiple-domain cognitive impairment (MDCI), as it was previously shown that individuals with MDCI with memory involvement have high rates of progression to Alzheimer’s disease [45], which is a major cause of morbidity and mortality in older individuals [44]. MDCI was assigned by the neuropsychology team under the direction of the study neuropsychologist. Annual neurocognitive batteries evaluating memory, language, visuospatial, attention, and executive cognitive domains were double scored and age normed. As controversy remains about optimal cut-off scores [46], performances 1.5 SD below the age-appropriate mean were defined as impaired [45]. Multiple-domain cognitive impairment assignment was made when participants had impaired performance on at least one measure of memory and impaired performance on at least one measure in another cognitive domain.

Composite incident morbidity was a composite outcome defined as onset of either cardiovascular disease, diabetes, cancer, or MDCI during study follow-up. To maximize power for the incident morbidity analysis, individuals with pre-existing disease were included in the analysis and were monitored for the onset of an additional age-associated morbidity. For the single disease analyses, individuals with that particular disease at baseline were excluded from the analysis (e.g., individuals with baseline diabetes were excluded from all analyses of incident diabetes but were included in the incident morbidity analysis).

### 2.4. Statistical Analysis

Statistical analysis was performed using custom scripts in Python (version 3.6), a general-purpose programming language. For all analyses tracking a particular biochemical measurement—IGF-1, IGFBP-1, IGFBP-3, or IGF-1/IGFBP-3 molar ratio—the participants were dichotomized into high and low groups using sex-specific medians of their baseline measurements. The median IGF-1 for men was 123 ng/mL and for women it was 105.5 ng/mL. Thus, the “low IGF-1” group was formed by combining men and women with IGF-1 levels below the sex-specific medians and the “high IGF-1” group was formed by combining men and women with IGF-1 levels at or above the sex-specific medians. “High” and “low” groups for analyses of IGFBP-1 (median 14 ng/mL in males, 18 ng/mL in females), IGFBP-3 (median 3.5 mg/L in males, 4.1 mg/L in females), and IGF-1/IGFBP-3 ratio (median 0.15 in males, 0.11 in females) were all formed by this procedure.

For comparison of baseline characteristics, normally distributed continuous variables (age and biochemical measurements) were compared using a two-tailed student’s *t*-test. Normality was confirmed by visual inspection of the histograms. Categorical variables were compared using either a chi-squared test of homogeneity (deaths) or a binomial test (number of participants). Results were considered statistically significant at *p*-value < 0.05.

Unadjusted survival curves for “high” and “low” groups of participants were generated using the Kaplan–Meier method for censored data and the survival curves were compared using log-rank tests. In addition, Cox proportional hazards models adjusted for sex and age at study enrollment were fit to the dichotomized biochemical measures and a clinical outcome of interest. The analyses were further stratified by sex, in order to identify any sex-specific differences in the association between dichotomized biochemical measures and clinical outcomes of interest. Interaction between age and dichotomized biochemical measures were investigated in mortality models stratified by median age at enrollment. Additionally, models were adjusted for dichotomized IGF-1 levels to investigate the independent associations between IGFBPs and clinical outcomes. BMI and insulin levels were included as covariates in models that predicted incidence of diabetes.

## 3. Results

### 3.1. Baseline Characteristics of Study Cohort

The study included 840 subjects (54.5% female), whose baseline characteristics are shown in Table 1 and in Appendix A. The median follow-up time for mortality was 6.9 years (interquartile range 4.6–8.5 years). The average age of the cohort was 76.1 ± 6.8 years, with no significant difference between men and women (*p* = 0.39). Men, on average, had higher serum IGF-1 levels (*p* < 0.001) and IGF-1/IGFBP-3 molar ratios (*p* < 0.001), but lower IGFBP-1 (*p* < 0.001) and IGFBP-3 (*p* < 0.001) compared to women. The average age was significantly older in subjects with low IGF-1 compared to those with high IGF-1 in the combined cohort (*p* = 0.04) and among men in the sex-stratified analysis (*p* = 0.02), (Appendix A). Over the course of follow-up, 13.9% of study participants died. At baseline, prevalence of morbidities was as follows: cardiovascular disease 12.7%, diabetes 10.5%, cancer 22.1%, and MDCI 3.0% (Appendix A).

### 3.2. IGF-Associated Proteins and Mortality: Low IGFBP-3 and High IGF-1/IGFBP-3 Molar Ratio Predict Mortality Risk

In unadjusted analysis, baseline IGF-1 levels were not predictive of mortality risk (Figure 1a–c). Upon adjustment for baseline age, we observed a non-significant trend towards higher mortality hazard with high IGF-1 levels in women (HR = 1.28, 95% CI 0.96–1.71, *p* = 0.09; Figure 2c). High levels of IGFBP-1, compared to low levels, were associated with significantly higher mortality risk in the overall cohort (*p* < 0.001) and among men (*p* < 0.001), (Figure 1d–f), but the associations became non-significant upon adjustment for age and sex (Figure 2a–c). On the other hand, high levels of IGFBP-3 predicted a lower mortality risk in an unadjusted analysis of the overall cohort (*p* < 0.001), as well as in men (*p* = 0.005) and women (*p* = 0.003) (Figure 1g–i). The difference in mortality risk between subjects with high vs. low IGFBP-3 remained significant upon adjusting for age and sex in the overall cohort (HR 0.82, 95% CI 0.680–0.998, *p* = 0.048), while in sex-stratified analysis associations retained the same direction but lost statistical significance (Figure 2a–c). Further adjustment for IGF-1 did not significantly alter the association between IGFBP-3 and mortality in the overall cohort (HR 0.71, 95% CI 0.56–0.89, *p* = 0.003) and strengthened it in women (HR 0.60, 95% CI 0.43–0.84, *p* = 0.003), but the association in men remained non-significant (HR 0.85, 95% CI 0.62–1.17 *p* = 0.32). High IGF-1/IGFBP-3 molar ratio, which is an estimate of circulating free IGF-1, was associated with higher mortality risk in the overall cohort (*p* = 0.002) and in women (*p* = 0.003) in unadjusted analysis (Figure 1j–l), and these associations persisted upon adjustment for age and sex (HR 1.28, 95% CI 1.05–1.57, *p* = 0.02 and HR 1.53, 95% CI 1.12–2.09, *p* = 0.007, respectively) (Figure 2a–c). After exclusion of 5 participants (4 males, 1 female) who died during the first year of follow-up, associations between IGF-1 and related proteins with mortality remained largely unchanged (data not shown).

Stratification by median participant baseline age showed consistent associations of IGF-related proteins and mortality between the two age groups (Appendix A), with the exception of IGFBP-1, which was significantly positively associated with mortality in younger (HR 1.60, 95% CI 1.04–2.46, *p* = 0.03) but not in older participants (HR 0.98, 95% CI 0.78–1.22, *p* = 0.85). High IGF-1, on the other hand, was more strongly associated with mortality among older women (HR 1.47, 95% CI 1.03–2.08, *p* = 0.03) compared with younger women (HR 1.10, 95% CI 0.64–1.87, *p* = 0.73).

### 3.3. IGF-Associated Proteins and Morbidity: High IGF-1 Predicts Risk for MDCI and Age-Related Composite Morbidity while Low IGFBP-1 Predicts Risk for Diabetes

High IGF-1 levels, compared to low IGF-1 levels, were associated with greater risk for incident MDCI in the overall cohort (*p* = 0.04) and in men (*p* = 0.045). These associations remained significant after adjusting for baseline age and sex, with HR 1.56, 95% CI 1.08–2.26, *p* = 0.02 and HR 1.81, 95% CI 1.04–3.16, *p* = 0.04 for MDCI in the overall cohort and in men, respectively. Similarly, we observed a greater risk with high IGF-1 for composite incident morbidity in the overall cohort (*p* = 0.04) and in men (*p* = 0.03) (Figure 3), which remained significant after adjustments (HR 1.242, 95% CI 1.004–1.538, *p* = 0.046 and HR 1.44, 95% CI 1.04–2.01, *p* = 0.03, respectively; Figure 4). High IGFBP-1, compared to low IGFBP-1 level, was also associated with higher risk for incident MDCI in men (*p* = 0.004), but not in the overall cohort (*p* = 0.11) or in women (*p* = 0.43), (Figure 5). After adjusting for age and sex, the association between IGFBP-1 level and MDCI hazard became non-significant (Figure 6). On the other hand, high IGFBP-1 was associated with reduced diabetes risk in the overall cohort in unadjusted analysis (*p* = 0.01), (Figure 5). In age- and sex-adjusted analysis, high IGFBP-1 remained significantly associated with protection from incident diabetes in the overall cohort (HR 0.50, 95% CI 0.29–0.88, *p* = 0.02) and in men (HR 0.31, 95% CI 0.10–0.92, *p* = 0.03; Figure 6). These associations persisted upon inclusion of IGF-1 as a covariate in the age- and sex-adjusted model (HR 0.50, 95% CI 0.29–0.89, *p* = 0.02 in the overall cohort; HR 0.30, 95% CI 0.10–0.89, *p* = 0.03 in men). However, when body mass index (BMI) and insulin levels were added to the model, the association between IGFBP-1 and diabetes was attenuated and no longer significant (HR 0.47, 95% CI 0.21–1.03, *p* = 0.06 in the overall cohort; HR 0.66, 95% CI 0.26–1.67, *p* = 0.38 in men). Levels of IGFBP-3 were not significantly associated with risk of any of the investigated age-related diseases (Appendix A). Associations between high IGF-1/IGFBP-3 ratio and MDCI risk were in the same directions as those between IGF-1 and MDCI, but they reached statistical significance only among women in both unadjusted (*p* = 0.04) and age-adjusted analyses (HR 1.81, 95% CI 1.03–3.21, *p* = 0.04), (Appendix A).

## 4. Discussion

In a longitudinal cohort of independently living older adults in generally good health, we found that IGF-1 and associated proteins predicted all-cause mortality and incidence of age-related diseases, including MDCI, diabetes, and composite incident morbidity. High IGF-1/IGFBP-3 molar ratio, which is considered an estimate of bioavailable IGF-1 [21], predicted a 28% greater risk of mortality, while high baseline IGF-1 level predicted a 56% greater risk for MDCI and a 24% greater risk for composite incident morbidity. The rationale for studying a morbidity composite that includes several major diseases is that aging is a risk factor for all age-related diseases. Therefore, a biological process that accelerates aging is expected to increase the risk for multiple age-associated diseases [47]. Our results confirm findings from model organisms [6,7,8] and cohorts with exceptional longevity [22,23,41], which demonstrated that attenuated IGF-1 levels or bioavailability were predictive of extended lifespan and health-span. While these findings are consistent with several other epidemiologic studies [48], we provide additional evidence for the role of GH/IGF-1 axis in mortality and morbidity, specifically among older adults. We have also shown that IGFBP-3 and IGFBP-1 predict mortality and diabetes, respectively. This contributes to the growing body of evidence that IGFBPs, in addition to their classical roles in regulating IGF-1 bioavailability, may also exert independent effects on lifespan and health-span.

The results from our study support the theory that diminished IGF-1 levels and bioavailability promote longevity and prolonged health-span in humans. The longevity-promoting mechanisms of diminished GH/IGF-1 signaling are well-studied in animal models and include improved stress defense, autophagy and cell survival via reduced PI3K/Akt and mTOR signaling [49,50]. Data from human studies have also shown that GH receptor deficiency improves defense from oxidative stress in healthy tissues and promotes apoptosis in neoplastic cells [17]. In genetic studies, mutations in the IGF-1 receptor that result in partial IGF-1 resistance [51] and polymorphisms in genes in insulin/IGF-1 signaling pathway [52,53], were associated with exceptional longevity. In exceptionally long-lived human cohorts, our group and others have shown that lower levels of IGF-1 and IGF-1/IGFBP-3 molar ratio predict longer survival [22,23], better cognitive function [41], and better functional status [23]. On the other hand, studies in older individuals who were not of exceptionally old age have shown inconsistent results. While one study in community-dwelling older adults found lower IGF-1 levels to be associated with decreased mortality [48], other studies in individuals with high cardiovascular risk found associations with increased mortality [24,25], or null results [26,38]. Our results offer additional evidence in support of lower IGF-1 levels being associated with reduced mortality and may bring us closer to resolving these inconsistencies.

It is important to note that the same IGF-1 level can represent different physiological states depending on the context and population studied. For instance, IGF-1 levels may be low due to an acute illness [31] or chronic disease [32], which could lead to findings of an inverse association between IGF-1 levels and mortality as a result of “reverse causation”. On the other hand, low IGF-1 level may reflect a lifelong diminished IGF-1 signaling due to genetic variants that may confer longevity. In fact, the relationship between IGF-1 levels and mortality may be bimodal in a heterogeneous population, as shown in a meta-analysis that included 12 studies and more than 14,000 subjects [54]. The association of low IGF-1 levels with mortality may reflect the presence of chronic disease, while the association of high IGF-1 levels with mortality might reflect life-long higher IGF-1 exposure. Our cohort was in good overall health, with relatively low prevalence of chronic diseases [55,56] and our findings were confirmed upon exclusion of those who died within the first year of follow-up. Therefore, the associations between IGF-1 levels and mortality were unlikely to be affected by pre-existing comorbid conditions and suggest protection by reduced IGF-1 signaling per se. The age of the cohort should also be taken into consideration. Since IGF-1 levels naturally decline with age [57], a low IGF-1 level in a younger individual may reflect an underlying disease or accelerated aging, whereas a low IGF-1 level in an older individual may reflect healthy physiology. Thus, age-interaction is important to consider in any analysis. Since our cohort was composed only of older individuals, it would not have been surprising not to find interactions between age and IGF-1-associated measures in prediction of mortality risk. However, even in this older cohort (mean age 76.0 for females) we detected a signal for greater hazard of mortality with higher IGF-1 in a subgroup of females above median age (mean age 81.8 years), which further supports the theory that high IGF-1 may be particularly detrimental in older individuals. In our cohort, low baseline IGF-1 bioavailability and levels predicted both delayed occurrence of age-associated morbidities and longer survival, supporting the notion that diminished IGF-1 signaling is associated with delayed aging. Furthermore, by conducting our analysis in a relatively healthy cohort of older age, we minimized many potential confounders.

Higher IGF-1 levels in our cohort were associated with incident MDCI. While the role of IGF-1 system in cognitive aging has been extensively studied, prior findings have not been conclusive [58]. Cross-sectional studies in middle-aged and older individuals have reported both positive [59,60] and negative [41,61] correlations between circulating IGF-1 levels and cognitive performance. Prospective studies have similarly shown conflicting findings. A study in older women, using phone-based neurocognitive assessments, found positive association between baseline IGF-1 and future cognitive performance [62]. The opposite was found in men of similar age in a study that used more comprehensive, in-person neurocognitive evaluations [61]. Some of these conflicting findings may be attributed to heterogeneity between the study populations, methods of cognitive assessment, and definitions of cognitive outcomes [58]. The comprehensive in-person neurocognitive assessments and diagnosis of MDCI, established by a neuropsychologist, increase confidence in the validity of our findings. Furthermore, the biological effects of IGF-1 on the brain may vary depending on age and type of insult [58]. IGF-1 promotes neurogenesis, synaptogenesis, myelination, and cell survival, which are important for brain development and repair after an acute injury [63,64]. On the other hand, IGF-1 increases oxidative stress and inhibits both autophagy and stress responses, leading to diminished cell resilience and accumulation of aberrant proteins and other cellular debris [49,65]. Consistent with these experimental data, interventional trials did not confirm protective cognitive effects of IGF-1 in older adults [66,67]. Presence of high levels of IGF-1 is therefore beneficial for the brain during youth and after an acute insult, but may be detrimental during aging and in evolving neurodegenerative diseases [58], which is supported by our results. As prior studies have shown that older individuals with MDCI with memory involvement have a high rate of conversion to Alzheimer’s disease [45], it will be important to further explore the role of IGF-1 in progression from cognitive impairment to Alzheimer’s disease in cohorts with larger number of participants and/or longer follow-up.

Our findings reaffirmed the negative association between IGFBP-3 and all-cause mortality previously noted by other studies [33,34,38]. Furthermore, we confirmed that the association between IGFBP-3 and all-cause mortality is independent from IGF-1 levels, as previously suggested [33,34,38]. These epidemiologic observations of IGFBP-3′s independent effects are supported by experimental evidence and possibly involve two different mechanisms. First, the functional nuclear localization sequence of IGFBP-3 allows it to enter the nucleus [68], where it has been shown to alter gene expression [69]. Second, IGFBP-3 may bind to a cell-surface receptor lipoprotein receptor-related protein-1 (LRP-1), which was shown to mediate inhibitory effects of IGFBP-3 on cellular growth [70]. Experimental [71,72] and epidemiologic studies [34] suggest that IGFBP-3 may exert its protective effects by reducing cancer-related mortality. However, not all studies have been consistent [73,74] and power limitations in our study precluded cause-specific mortality analysis.

While low levels of IGFBP-1 have been associated with increased risk for diabetes in middle-aged individuals [75,76,77], our study is among the first to show that low IGFBP-1 levels may predict diabetes in older adults. Circulating IGFBP-1, which is produced mainly by the liver, is normally suppressed in postprandial state by hyperinsulinemia [78] and increased glycolysis [79]. As the levels of IGFBP-1 fluctuate throughout the day in response to feeding and fasting, it acutely regulates the availability of free IGF-1, which has insulin-sensitizing effects [80]. In adipose tissue, IGFBP-1 inhibits IGF-1 stimulated proliferation of preadipocytes [81], resulting in reduced fat mass [82]. At the same time, IGFBP-1 may promote insulin secretion and glucose uptake independently of IGF-1, via binding to a cell-surface integrin receptor [83]. In line with these experimental findings, several cross-sectional and prospective epidemiologic studies have associated low IGFBP-1 levels with obesity [84], high fasting insulin [75], impaired glucose tolerance, and diabetes [75,76,85] in middle-aged individuals. However, prospective data on the association between IGFBP-1 and diabetes risk in older adults are scarce. In our cohort of older adults, we found in an age-adjusted analysis that low IGFBP-1 predicted risk for diabetes. This association persisted after adjusting for IGF-1 but was attenuated with inclusion of BMI and insulin in the model. These findings indicate that the protective effects of high IGFBP-1 against diabetes may be partly mediated by lower BMI and related enhanced insulin sensitivity in individuals with high IGFBP-1.

The sex-stratified analysis highlighted that some of the studied associations may be sex-specific or preferential. For instance, we found that IGF-1/IGFBP-3 molar ratio may be a better predictor of mortality in women than in men, similar to the findings from many rodent [6,10,11] and human [22] studies. It is well known that levels of various elements of the GH/IGF-1 system vary between men and women. Men have higher levels of IGF-1 and IGF-1/IGFBP-3 molar ratio, whereas women have higher levels of IGFBP-3 and IGFBP-1 [21,86]. Women also have diminished physiologic response to GH, which translates into lower IGF-1 levels and alterations in body water and fat content [87]. However, it is not established whether these differences contribute to observed divergence in mortality and morbidity between men and women. Future studies are needed to clarify if hormonal or other sex-specific factors interact with signaling in the GH/IGF-1 pathway.

Although our study possesses many unique strengths, it also has some limitations. Assessment of GH/IGF-1 pathway activity in humans remains challenging due to the inherent complexity of this biological system [36]. Additionally, it has been noted that total IGF-1 level may be an imperfect proxy for bioavailable IGF-1 [88]. However, the high-affinity with which IGF-1 binds to IGFBPs has limited the development of a reliable laboratory assay for measuring free IGF-1 [37]. We therefore used IGF-1/IGFBP-3 molar ratio as an estimate of free IGF-1, similar to a number of previous studies [23,89,90,91]. IGF-1/IGFBP-3 ratio has been shown to positively correlate with free IGF-1 [92] and has been associated with a number of clinical outcomes, including functional status in nonagenarians [23], metabolic disease [89,90], and neoplastic diseases [91,93]. Regardless of the selected measure, a single measure of IGF-1 level and its associated proteins does not capture adequately the life-long exposure to IGF-1; thus, a longitudinal study with repeated measures would be needed to investigate the role of IGF-1 trajectories in human longevity and health-span. Another important member of the GH/IGF-1 system and the most abundant IGF in circulation is IGF-2. IGF-2, which has been implicated in disease [91,94], binds to the same IGFBPs and receptors as IGF-1, although at lower affinity, and signals via shared pathways with IGF-1 [95]; however, assessment of IGF-2 levels was out of scope of this study. Additionally, due to general good health of our cohort [55,56], there were relatively few incident disease events, which limited our power to study some of the age-related disease outcomes, in particular in sex-stratified analyses. However, the fact that our cohort was in good health allowed us to interpret our findings more reliably in the context of healthy aging and to minimize confounding that may arise from alterations in the GH/IGF-1 axis as a result of disease. Furthermore, all study samples were collected in the morning and under fasting conditions. This was particularly relevant for the interpretation of IGFBP-1 levels, which normally fluctuate in relation to prandial status; yet, not all prior studies have been able to establish these conditions [39].

In conclusion, our findings indicate that higher IGF-1 levels and/or bioavailability are predictive of mortality and morbidity risk. These results support the hypothesis that diminished signaling via GH/IGF-1 pathway may contribute to longevity and health-span in humans. If the detrimental effects of high IGF-1 signaling in older adults are confirmed by larger studies with longer follow-up time, then the GH/IGF-1 pathway may represent a promising target for therapies that delay aging. A monoclonal antibody that targets IGF-1 receptor (IGF-1R) and decreases IGF-1 signaling has already been shown to increase health-span and lifespan in middle-age female mice [96]. In fact, several FDA approved drugs that inhibit GH/IGF-1 signaling are currently in clinical use for other indications. For example, pegvisomant, a growth hormone receptor antagonist, is used for normalizing IGF-1 levels in acromegaly [97] and teprotumumab, which antagonizes IGF-1R, is used to treat thyroid eye disease [98]. These drugs could be readily repurposed for slowing aging in clinical trials. The findings in our study highlight the relevance of this evolutionarily conserved longevity pathway in human aging, and underscore the importance of future studies. In particular, investigating the longitudinal trajectories of circulating IGF-1 and associated proteins and genetically quantifying GH/IGF-1 signaling could serve to strengthen the causal connection between the GH/IGF-1 pathway and human aging.

## Figures and Tables

**Figure 1 cells-09-01368-f001:**
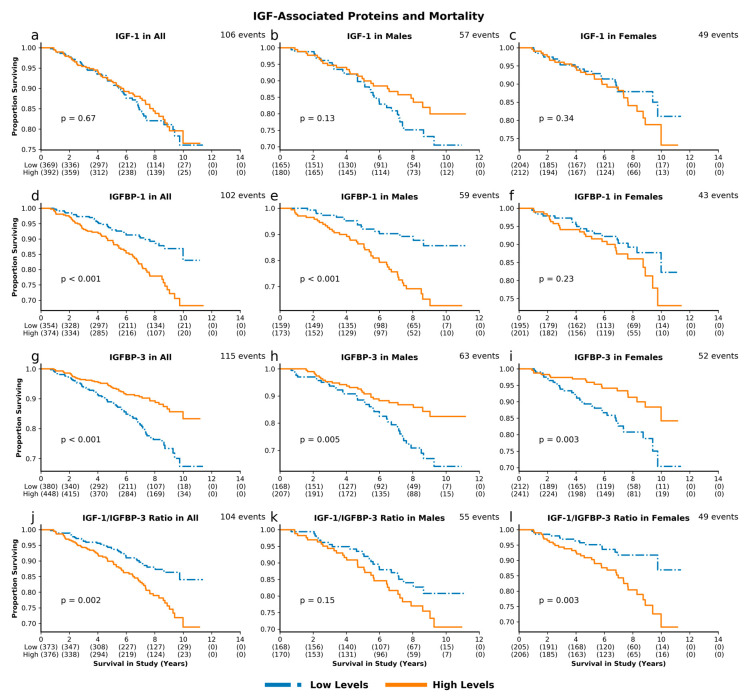
Insulin-like growth factor (IGF)-associated proteins and mortality. Unadjusted survival curves for individuals with high and low levels of IGF-1 (**a**. combined cohort; **b**. males; **c**. females), IGFBP-1 (**d**–**f**), IGFBP-3 (**g**–**i**), and IGF-1/IGFBP-3 molar ratio (**j**–**l**).

**Figure 2 cells-09-01368-f002:**
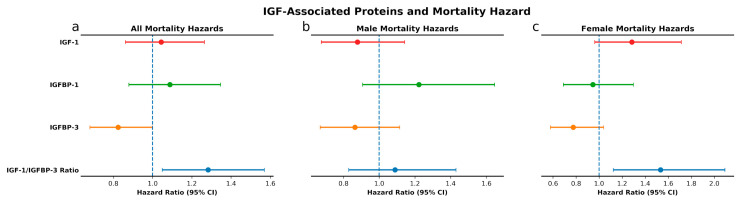
IGF-associated proteins and mortality hazard. Sex and age-adjusted survival hazards for combined cohort (**a**) and age-adjusted survival hazard for males (**b**) and females (**c**) with high levels of IGF-associated proteins as compared to individuals with low levels.

**Figure 3 cells-09-01368-f003:**
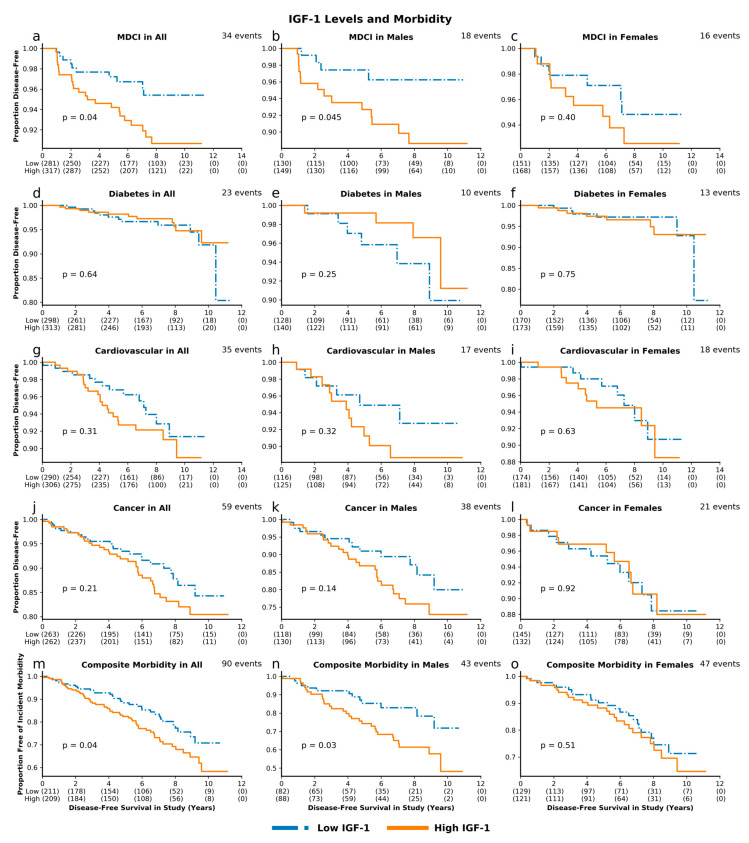
IGF-1 levels and morbidity. Unadjusted survival curves for multiple-domain cognitive impairment (MDCI) (**a**. combined cohort; **b**. males; **c**. females), diabetes (**d**–**f**), cardiovascular disease (**g**–**i**), cancer (**j**–**l**), and composite incident morbidity (**m**–**o**) in individuals with high and low levels of IGF-1.

**Figure 4 cells-09-01368-f004:**
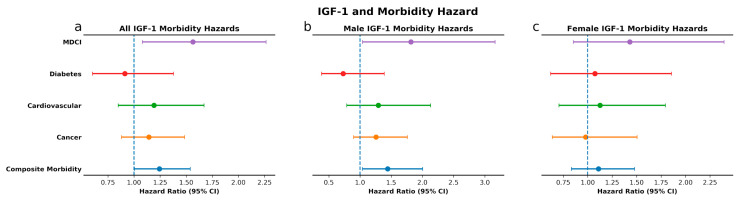
IGF-1 and morbidity hazard. Sex and age-adjusted morbidity hazards for all individuals in cohort (**a**), and age-adjusted morbidity hazards for males (**b**) and females (**c**) with high levels of IGF-1 as compared to individuals with low levels.

**Figure 5 cells-09-01368-f005:**
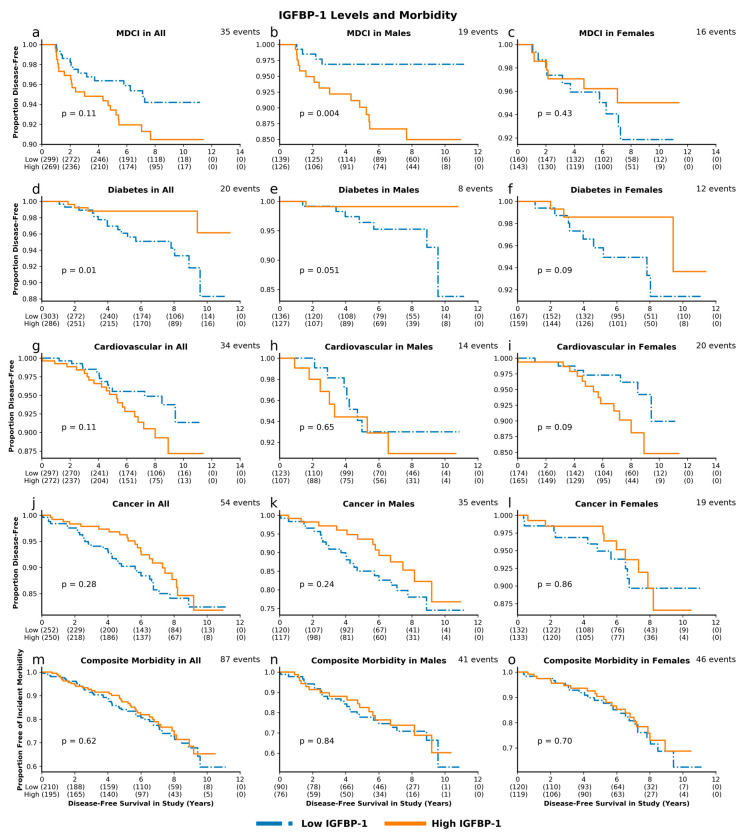
IGFBP-1 levels and morbidity. Unadjusted survival curves for MDCI (**a**. combined cohort; **b**. males; **c**. females), diabetes (**d**–**f**), cardiovascular disease (**g**–**i**), cancer (**j**–**l**), and composite incident morbidity (**m**–**o**) in individuals with high and low levels of IGFBP-1.

**Figure 6 cells-09-01368-f006:**
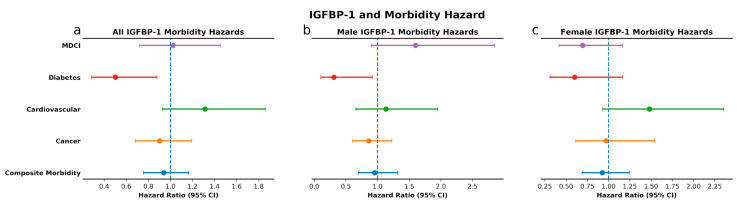
IGFBP-1 and morbidity hazard. Sex and age-adjusted morbidity hazards for all individuals (**a**), and age-adjusted morbidity hazard for males (**b**) and females (**c**) with high levels of IGFBP-1 as compared to individuals with low levels.

**Table 1 cells-09-01368-t001:** Baseline characteristics of study cohort. All *p*-values are for comparisons between males and females.

	All	Male	Female	*p*-Value
**Number of Individuals, *n* (%)**	840	382 (45.5)	458 (54.5)	0.01
**Deaths, *n* (%)**	117 (13.9)	65 (17.0)	52 (11.4)	0.02
**Age (years), mean ± SD**	76.1 ± 6.8	76.4 ± 7.0	76.0 ± 6.7	0.39
**BMI (kg/m^2^), mean ± SD, *n* = 820**	27.6 ± 4.7	27.9 ± 3.9	27.3 ± 5.3	0.053
**Insulin (mIU/L), mean ± SD, *n* = 801**	15.4 ± 12.3	16.6 ± 15.7	14.5 ± 8.3	0.02
**IGF-1 (ng/mL), mean ± SD, *n* = 761**	117 ± 38	127 ± 39	108 ± 36	<0.001
**IGFBP-1 (ng/mL), mean ± SD, *n* = 728**	19 ± 15	17 ± 14	21 ± 15	<0.001
**IGFBP-3 (mg/L), mean ± SD, *n* = 828**	3.9 ± 1.0	3.6 ± 0.9	4.2 ± 1.0	<0.001
**IGF-1/IGFBP-3 Molar Ratio, *n* = 749**	0.13 ± 0.04	0.15 ± 0.04	0.11 ± 0.03	<0.001

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
