# Peer review of "Insulin-like Growth Factor-1 and IGF Binding Proteins Predict All-Cause Mortality and Morbidity in Older Adults"

_cells, 2020, doi:10.3390/cells9061368_

Round 1

Reviewer 1 Report

William B. Zhang et al. show the role of insulin-like growth factor-1 and IGF-1 Binding Proteins related to predictions on mortality and morbidity. Overall, the manuscript is relevant to the medical field and is well presented. Some minor concerns should be considered:

  1. Start the abstract with an opening phrase: The growth hormone/insulin-like growth factor-1 (GH/IGF-1) pathway plays essential roles in growth, and development. Diminished...
  2. It should be helpful to discuss the results found for future advances in medicine or pharmacology, i.e. how it would be possible through some drugs (or if described already in animal models) to decrease IGF/IGFp levels to improve health...  

Reviewer 2 Report

This manuscript is well designed and has a simple but rational conclusion. I would like to recommend acceptance in the present form.

Reviewer 3 Report

The paper is well written, and the methodology and analysis are sound, with a unique population cohort. The follow-up time is relatively short (4.6-8.5 years) and, in reality, this is a study of IGF-I, IGFBP-1 and IGFBP-3 in older healthy subjects, as predictors of morbidity and mortality. Concentrations of each of these change with normal aging and indeed adjusting for age altered the predicted risk, abolishing the association of IGF-I and IGFBP-1 with mortality. The importance of the age interaction for IGF-I is well discussed in the discussion.

I have a few points that should be acknowledged in a revised manuscript:

The abbreviation IGFBP is for IGF-binding protein. In presenting these as IGF-I binders only, the important role of IGF-II in human health and disease is overlooked. It is surprising to me that there is no mention of IGF-II in this manuscript. At the very least, its existence as the most abundant IGF should be acknowledged.

IGF-I is expressed as a ratio to IGFBP-3. Why not introduce IGFBP-3 values into the statistical model instead? There are six IGFBPs with multiple post-translational forms. Low molecular weight species increase with age, and in a variety of disease states. Use of a single ratio such as this might be misleading and should be justified: what is the relationship between this ratio and more precise measures of free IGFs or readily available IGFs in this population? At the very least, the complexity of the IGF system should be acknowledged in the discussion.

The association of IGFBP-1, and not IGF-I with diabetes risk is well documented in earlier studies that could have been cited e.g. Diabetologia 2008, 51, 1135–1145. Since insulin is an important transcriptional inhibitor of IGFBP-1, it would be useful to present the fasting insulin values alongside those of IGFBP-1, if these are available.
